# Ceftazidime resistance in *Pseudomonas aeruginosa* is multigenic and complex

**Kay A. Ramsay**[1¤]*, **Attika Rehman**[1©], **Samuel T. Wardell**[1,2©], **Lois W. Martin**[1], **Scott C. Bell**[3,4], **Wayne M. Patrick**[5], **Craig Winstanley**[6], **Iain L. Lamont**[1]*

1 Department of Biochemistry, University of Otago, Dunedin, New Zealand, 2 Department of Microbiology and Immunology, University of Otago, Dunedin, New Zealand, 3 Department of Thoracic Medicine, The Prince Charles Hospital, Chermside, Queensland, Australia, 4 Children's Health Research Centre, Faculty of Medicine, The University of Queensland, South Brisbane, Queensland, Australia, 5 School of Biological Sciences, Victoria University of Wellington, Wellington, New Zealand, 6 Department of Clinical Infection, Microbiology and Immunology, Institute of Infection, Veterinary and Ecological Sciences, University of Liverpool, Liverpool, United Kingdom

© These authors contributed equally to this work.
¤ Current address: Child Health Research Centre, The University of Queensland, South Brisbane, Queensland, Australia
* k.ramsay@uq.edu.au (KAR); iain.lamont@otago.ac.nz (ILL)

**Data Availability Statement:** All relevant data are within the manuscript and its Supporting Information files.

## Abstract

*Pseudomonas aeruginosa* causes a wide range of severe infections. Ceftazidime, a cephalosporin, is a key antibiotic for treating infections but a significant proportion of isolates are ceftazidime-resistant. The aim of this research was to identify mutations that contribute to resistance, and to quantify the impacts of individual mutations and mutation combinations. Thirty-five mutants with reduced susceptibility to ceftazidime were evolved from two antibiotic-sensitive *P. aeruginosa* reference strains PAO1 and PA14. Mutations were identified by whole genome sequencing. The evolved mutants tolerated ceftazidime at concentrations between 4 and 1000 times that of the parental bacteria, with most mutants being ceftazidime resistant (minimum inhibitory concentration [MIC] ≥ 32 mg/L). Many mutants were also resistant to meropenem, a carbapenem antibiotic. Twenty-eight genes were mutated in multiple mutants, with *dacB* and *mpl* being the most frequently mutated. Mutations in six key genes were engineered into the genome of strain PAO1 individually and in combinations. A *dacB* mutation by itself increased the ceftazidime MIC by 16-fold although the mutant bacteria remained ceftazidime sensitive (MIC < 32 mg/L). Mutations in *ampC*, *mexR*, *nalC* or *nalD* increased the MIC by 2- to 4-fold. The MIC of a *dacB* mutant was increased when combined with a mutation in *ampC*, rendering the bacteria resistant, whereas other mutation combinations did not increase the MIC above those of single mutants. To determine the clinical relevance of mutations identified through experimental evolution, 173 ceftazidime-resistant and 166 sensitive clinical isolates were analysed for the presence of sequence variants that likely alter function of resistance-associated genes. *dacB* and *ampC* sequence variants occur most frequently in both resistant and sensitive clinical isolates. Our findings quantify the individual and combinatorial effects of mutations in different genes on ceftazidime susceptibility and demonstrate that the genetic basis of ceftazidime resistance is complex and multifactorial.

**Funding:** Authors who received the award: IL, WP, SB, CW Grant number: 17/372 Full name of funder: Health Research Council of New Zealand URL: www.hrc.govt.nz The funder had no role in study design, data collection and analysis, decision to publish, or preparation of the manuscript.

**Competing interests:** The authors have declared that no competing interests exist.

## Introduction

*Pseudomonas aeruginosa* is an opportunistic pathogen that is found in a variety of environments and causes acute and chronic infections in patients with a wide range of predisposing conditions [1]. The high frequency of *P. aeruginosa* infections coupled to its propensity to develop antibiotic resistance has caused it to be classified as one of the six most problematic bacterial pathogens [2] with resistant *P. aeruginosa* being estimated to be associated with over 250,000 deaths worldwide in 2019 [3]. As with other species, the time required for conventional antibiotic susceptibility testing can delay effective treatment, negatively impacting on patient outcomes [4, 5]. This has led to efforts to predict antibiotic effectiveness more rapidly by analysing the genomes of infecting bacteria [6–8] but this approach requires understanding of the genetic basis of resistance.

Ceftazidime, a cephalosporin antibiotic, is used to treat a wide range of *P. aeruginosa* infections including urinary tract infections, hospital-acquired pneumonia and ventilator-associated infections, and abdominal and invasive infections [9, 10]. However ceftazidime resistance has emerged with high frequencies in many settings including neonatal and pediatric infections [11], intensive care units [12], and burns wards [13]. Isolates that are resistant to cephalosporins are also frequently resistant to carbapenem antibiotics such as meropenem [14, 15].

Ceftazidime resistance is associated with overexpression of the MexAB-OprM efflux pump due to mutations in the *nalC*, *nalD* or *mexR* regulatory genes [15–19]. It can also involve increased expression of AmpC ß-lactamase due to mutations in *dacB* that encodes penicillin-binding protein 4 (PBP4), *ampD* that encodes an enzyme involved in peptidoglycan recycling, and *ampR* that encodes a regulator of *ampC* expression. Mutations in *ampC* itself can also increase activity of the AmpC enzyme against ceftazidime [20]. *In vitro* studies, experimentally evolving ceftazidime-resistant mutants from the sensitive reference strains PAO1 [18, 21] or PA14 [22], resulted in frequent occurrence of mutations in *dacB* (PBP4) and, less frequently, *mpl* that are involved in regulating AmpC production [23–26]. Mutations also arose in genes encoding regulators of MexAB-OprM synthesis, consistent with the contributions of this efflux pump to ceftazidime resistance. Additionally, some resistant mutants had large (up to 443 kb) genome deletions that may contribute to resistance through loss of the *galU* gene that is involved in lipopolysaccharide (LPS) synthesis [22]. Many mutants selected for ceftazidime resistance were also resistant to carbapenems [21, 22]. Mutants engineered to lack genes encoding peptidoglycan remodelling enzymes also had changes to ß-lactam susceptibility [27].

Although genetic mechanisms associated with ceftazidime resistance have been identified and characterised, the interplay between different mechanisms and the effects of combinations of naturally occurring mutations have not been explored. The aims of this research were to develop a large number of experimentally evolved mutants to develop a deeper understanding of the genes and mutations that contribute to resistance, and to use that information to engineer mutant combinations and hence quantify the contributions of mutations in key genes, and of gene-gene interactions, to ceftazidime resistance.

## Materials and methods

### Experimental evolution of ceftazidime-resistant mutants

A gradient plate methodology was used to evolve mutants of *P. aeruginosa* resistant to increasing concentrations of ceftazidime on Mueller-Hinton (MH) agar, as described previously [28]. Briefly, overnight cultures of bacteria grown in Luria-Bertani (LB) broth were diluted to 1.5 x $10^6$ cfu/mL, inoculated onto agar with a gradient of ceftazidime across the plate, and incubated at 37°C for at least18 hours. From each plate, one colony growing at the highest concentration

of ceftazidime was inoculated into antibiotic-free LB broth and the resulting culture inoculated onto a gradient plate with a two-fold increase in the concentration of ceftazidime. This cycle was repeated until mutants with increased resistance were not obtained. All mutants were from independent starter cultures.

### Minimum inhibitory concentration (MIC) determination

The MIC was determined using the doubling dilution method on agar plates [28, 29], with at least three biological replicates for each mutant. Bacteria were categorised into resistant and sensitive/intermediate phenotypes following CLSI guidelines [30]. Ceftazidime and meropenem have resistance breakpoints of $\geq$ 32 mg/L and $\geq$ 8 mg/L respectively.

### Whole genome sequencing and bioinformatic analysis

Genome sequencing was performed by either the Otago Genomic Facility or Custom Science using the Illumina platform, with read lengths of 300 bp with approximately 30x coverage or 150 bp with approximately 50x coverage, respectively. Following processing of raw reads, they were mapped to the parental bacteria (PAO1-Otago or PA14-Otago) to identify mutations using BreSeq version 0.35.0 [28, 31]. The genomic differences between the PAO1_Otago and PA14_Otago genomes and refseq reference genome sequences (NC 002516 and NC 008463) have been described [28, 32]. The likely effects of gene sequence variants on protein function were analysed using PROVEAN [28, 33]. Genomes were analysed for the presence of acquired ß-lactamases using ResFinder v. 4.2 [34].

### Allelic exchange for the construction of *P. aeruginosa* mutants

Mutations were engineered in reference strain PAO1using a two-step allelic exchange method [35] as described previously [36]. Briefly, mutation-containing regions of interest were amplified by PCR using appropriate primers (S1 Table in S1 File) from mutants evolved in this or a previous study [28] and cloned into plasmid pEX18Tc [37]. The resulting plasmids were sequenced to confirm that only the expected mutation was present, transformed into *E.coli* strain ST18 [38], and transferred into strain PAO1 by biparental mating, allowing the mutations to be transferred into the *P. aeruginosa* genome. PCR amplicon sequencing was carried out to identify mutant bacteria.

## Results

### Experimental evolution of ceftazidime-resistant mutants

Ceftazidime-resistant mutants of *P. aeruginosa* were selected using multiple cycles of agar gradient plate selection interspersed with growth in antibiotic-free broth. Twenty-five resistant mutants were derived from reference strain PAO1 or a *lacZ*-marked derivative. The MICs for the mutants ranged from 4 to 1024 mg/L of ceftazidime (median 128 mg/L), up to 1000-fold higher than the parental PAO1 strain (1 mg/L). To determine whether the genetic background caused significant changes in resistance mechanisms, ten further mutants were derived from reference strain PA14 (MIC 1 mg/L) and had MICs of between 16 and 1024 mg/L (median 512 mg/L). *P. aeruginosa* are considered to be clinically resistant to ceftazidime if the MIC is $\geq$ 32 mg/L [30]. MIC values of individual mutants are listed in S2 and S3 Tables in S1 File.

Ceftazidime-resistant *P. aeruginosa* can also have reduced susceptibility to meropenem, a carbapenem antibiotic [21]. Meropenem MICs of the mutant bacteria ranged from 2 to 16 mg/L for mutants derived from strain PAO1 (median, 4 mg/L), higher than the parental strain (1 mg/L); and from 2 to 32 mg/L for mutants derived from strain PA14 (median, 16 mg/L

whereas the parental strain had an MIC of 0.5 mg/L) (S2 and S3 Tables in S1 File). An MIC of ≥ 8 mg/L of meropenem is classified as being resistant [30]. As well as being ceftazidime-resistant, ten of the ceftazidime-selected PAO1 mutants and nine of the PA14 mutants were therefore also resistant to meropenem.

### Resistance-associated mutations

Whole genome sequencing of the resistant mutants showed that the PAO1 mutants had between 1 and 10 mutations each (median 6), while the PA14 mutants had between 5 and 10 mutations (median 8). Genes that were mutated in two or more mutants are listed in Tables 1 and 2. Mutated genes largely fell into five categories, encoding peptidoglycan-related enzymes, regulators of efflux, enzymes involved in LPS biosynthesis, enzymes involved in metabolic processes and enzymes involved in synthesis of macromolecules. In addition, a high proportion (13/35) of resistant mutants had large (up to 925 kb) genome deletions. Only 2 of the identified changes were synonymous (silent) indicating a strong selection pressure for non-synonymous changes.

In the 25 PAO1 mutants, a total of 53 genes contained mutations, with 21 genes being mutated in two or more mutants (Table 1, S2 Table in S1 File). Mutations in at least one of the *dacB* and *mpl* genes occurred in 16 of the mutants. Mutations in these genes alter processing and recycling of peptidoglycan resulting in increased expression of the *ampC* ß-lactamase that has low-level enzymatic activity as a ceftazidime hydrolase [23–26]. Three mutants had mutations in *sltB1* that is also involved in peptidoglycan processing [27, 39]. Seven mutants had mutations in *ampC* itself, likely increasing the activity of the enzyme against the ceftazidime substrate [20, 40].

Mutations in regulatory proteins can also contribute to antibiotic resistance. Five of the ceftazidime-resistant mutants had mutations in *nalD* that encodes a regulator of the MexA-B-OprM efflux pump. An additional three mutants had mutations in the gene encoding the two-component sensor protein PhoQ involved in controlling expression of multiple genes associated with antibiotic resistance [41].

Over half of the mutants had mutations in genes encoding enzymes involved in LPS synthesis. So far as we are aware, mutations affecting LPS biosynthesis have not previously been reported to contribute to ceftazidime resistance but their independent occurrence in multiple mutants indicates that they are associated with tolerance to higher concentrations of ceftazidime. A high proportion of the mutants also had mutations affecting proteins involved in amino acid, nucleotide or fatty acid synthesis and in genes affecting nucleic acid and protein synthesis. Very strikingly, seven of the evolved PAO1 mutants had large (>100 kb) deletions. The largest was 925 kb (14.8% of the genome, 748 genes), extending from PA1818 (*cadA*) to PA2566. The other large deletions were all within the same region of the genome (S2 Table in S1 File). The occurrence of large deletions has been observed previously in *P. aeruginosa* mutants experimentally evolved to be resistant to ceftazidime or meropenem [21, 22, 28, 42].

The ten experimentally evolved ceftazidime-resistant mutants of strain PA14 also had a wide range of mutations. A total of 38 genes were mutated, with 14 genes being mutated in two or more mutants (Table 2, S3 Table in S1 File). Many of the mutated genes contribute to the same cellular processes as mutated genes in strain PAO1 mutants. These included *ampC* and genes involved in regulation of its expression and genes for MexAB-OprM efflux pump regulation. Furthermore, three of the PA14 mutants had mutations in *mexB* that encodes a structural component of the MexAB-OprM efflux pump. Genes involved in LPS synthesis were also mutated in 4 of the 10 mutants.

Eight of the PA14 mutants had mutations in genes involved in a range of metabolic processes including respiration, fatty acid synthesis and carbon metabolism, and synthesis of

**Table 1. Genetic variants following ceftazidime experimental evolution in *Pseudomonas aeruginosa* strain PAO1 and PAO1::*lacZ* (n = 25).**

| Gene category/ name | Locus Tag | Function if known | Prevalence of mutations* |
|---|---|---|---|
| **AmpC ß-lactamase and peptidoglycan processing** | | | |
| *dacB* | PA3047 | PBP4, D-alanyl-D-alanine carboxypeptidase | 15 |
| *mpl* | PA4020 | UDP-N-acetylmuramate:L-alanyl-gamma-D-glutamyl-meso-diaminopimelate ligase | 12 |
| *ampC* | PA4110 | ß-lactamase precursor | 7 |
| *sltB1* | PA4001 | soluble lytic transglycosylase B | 3 |
| **MexAB-OprM efflux pump and its regulation** | | | |
| *nalD* | PA3574 | NalD repressor | 5 |
| **Lipopolysaccharide synthesis** | | | |
| *wapH* | PA5004 | WapH | 10 |
| *wbpL* | PA3145 | glycosyltransferase WbpL | 3 |
| *ssg* | PA5001 | cell surface-sugar biosynthetic glycosyltransferase, SsgAdd | 2 |
| **Metabolic processes** | | | |
| PA4333 | PA4333 | probable fumarase | 5 |
| PA3172 | PA3172 | probable hydrolase | 2 |
| *ilvC* | PA4694 | ketol-acid reductoisomerase | 2 |
| **Synthesis and processing of macromolecules** | | | |
| *infA* | PA2619 | translation initiation factor IF-1 | 5 |
| *clpA* | PA2620 | ATP-dependent Clp protease, ATP-binding subunit ClpA | 2 |
| *spoT* | PA5338 | guanosine-3',5'-bis(diphosphate) 3'-pyrophosphohydrolase | 2 |
| *rpsA* | PA3162 | 30S ribosomal protein S1 | 2 |
| **Other functions/ unknown** | | | |
| Large deletion† | | | 7 |
| PA1767 | PA1767 | hypothetical protein | 4 |
| *phoQ* | PA1180 | two-component sensor PhoQ | 3 |
| *pilP* | PA5041 | type 4 fimbrial biogenesis protein PilP | 2 |
| PA2982 | PA2982 | Conserved hypothetical protein | 2 |
| *vacJ* | PA2800 | VacJ | 2 |

*Genes mutated in two or more mutants are listed. A complete list of mutations is in S2 Table in S1 File.

†Deletions of 173–925 kb. One mutant also had a second deletion of 36 kb. Deleted genes are listed in S2 Table in S1 File.

amino acids, pyrimidines or enzyme cofactors. In contrast to strain PAO1, only one mutation, in the *spoT* gene, is likely to directly synthesis of macromolecules. Conversely, a higher proportion of the PA14 mutants than the PAO1 mutants had mutations in *phoQ* and the protease-encoding *clpA* or *clpS* genes.

Like PAO1, a significant proportion (6 out of 10) of the PA14 mutants had large deletions (194–460 kb). These deletions were in the corresponding part of the genome to the deletions in strain PAO1 (spanning the PA14 genes equivalent to PA1893 to PA2572) (S3 Table in S1 File).

## Effects of individual mutations on MIC

To determine the effects of individual mutations on antibiotic susceptibility, mutations were engineered into the genome of strain PAO1. Six genes, with a total of eight genetic variants, were chosen based on their frequencies of mutation in the experimentally-evolved mutants and potential contributions to resistance described in the scientific literature. They were *dacB*

**Table 2. Genetic variants identified following ceftazidime experimental evolution studies in *Pseudomonas aeruginosa* strain PA14 (n = 10).**

| Gene category/ name | Locus Tag | Function if known | Prevalence of mutations* |
|---|---|---|---|
| **AmpC ß-lactamase and peptidoglycan processing** | | | |
| *dacB* | PA14_24690 | PBP4, D-alanyl-D-alanine carboxypeptidase | 4 |
| *mpl* | PA14_11845 | UDP-N-acetylmuramate:L-alanyl-gamma-D-glutamyl meso-diaminopimelate ligase | 6 |
| *ampR* | PA14_10800 | transcriptional regulator AmpR | 2 |
| **MexAB-OprM efflux pump and its regulation** | | | |
| *nalD* | PA14_18080 | TetR family transcriptional regulator | 3 |
| *mexB* | PA14_05540 | RND multidrug efflux transporter MexB | 3 |
| *nalC* | PA14_16280 | transcriptional regulator | 2 |
| **Lipopolysaccharide synthesis** | | | |
| *orfN* (*wbpL* in PAO1) | PA14_23460 | group 4 glycosyl transferase | 4 |
| **Metabolic processes** | | | |
| *pgi* | PA14_62620 | glucose-6-phosphate isomerase | 2 |
| **Synthesis and processing of macromolecules** | | | |
| *clpA* | PA14_30230 | ATP-dependent Clp protease, ATP-binding subunit ClpA | 6 |
| **Other functions/ unknown** | | | |
| Large deletion† | | | 6 |
| *hepP* | PA14_23430 | HepP | 3 |
| *minC* | PA14_22040 | septum formation inhibitor | 2 |
| *phoQ* | PA14_49170 | two-component sensor PhoQ | 8 |
| PA14_55770 | PA14_55770 | phosphate transporter | 2 |
| *tatC* | PA14_66980 | sec-independent protein translocase TatC | 2 |

*Genes mutated in two or more mutants are listed. A complete list of mutations is in S3 Table in S1 File.

†Deletions of 194–460 kb. One mutant had two deletions of 320 and 460 kb. Deleted genes are listed in S3 Table in S1 File.

(13bp deletion), *mpl* (1 bp insertion), *ampC* (amino acid substitutions P180L and G242R), *nalD* (amino acid substitutions T11N and T158P), all of which arose frequently in the experimentally evolved mutants, and *nalC* (1 bp insertion) and *mexR* (Q128*) that were infrequently mutated in the evolved mutants but have been previously associated with ß-lactam resistance [16, 43, 44]. The 13bp deletion in *dacB* caused a 16-fold increase in the MIC for ceftazidime, relative to the parental strain (Table 3). All other mutations increased the ceftazidime MIC by 2- or 4-fold, except for the *mpl* mutation that had no effect. The MIC of meropenem was increased 2-fold by *nalC*, *nalD*$_{T11N}$ and *mexR* mutations and was not affected by the other mutations (Table 3).

MICs were also determined for mutants with transposon insertions in the genes under study (Table 4). The MICs were very similar to those of the engineered mutants, with a *dacB*::*Tn* mutation resulting in a 16-fold increase in the ceftazidime MIC, mutations in *mexR* and *nalD* mutation giving smaller increases and *mpl*::*Tn* mutations having no effect on the MIC. Only the *nalD* and *mexR* transposon mutations increased the MIC for meropenem. Transposon insertions in *ampC* did not significantly affect the MIC.

## Only some mutation combinations increase the MIC

Next we engineered combinations of mutations in *dacB*, *mpl*, *ampC* and *nalD*, four of the most frequently mutated genes in experimentally evolved PAO1 mutants, on MICs. Double

**Table 3. Ceftazidime and meropenem MIC for engineered mutants.**

| Mutations | Ceftazidime MIC† | Meropenem MIC |
|---|---|---|
| None (wild-type PAO1) | 1 | 1 |
| $dacB_{\Delta 13}$ | 16 | 1 |
| $mpl_{insG}$ | 1 | 1 |
| $nalC_{insC}$ | 4 | 2 |
| $nalD_{T11N}$ | 4 | 2 |
| $nalD_{T158P}$ | 2 | 1 |
| $ampC_{P180L}$ | 2 | 1 |
| $ampC_{G242R}$ | 2 | 1 |
| $mexR_{Q128*}$ | 4 | 2 |
| $dacB_{\Delta 13}\ ampC_{P180L}$ | 32 | 0.5 |
| $dacB_{\Delta 13}\ ampC_{G242R}$ | 64 | 1 |
| $dacB_{\Delta 13}\ mpl_G$ | 32 | 1 |
| $dacB_{\Delta 13}\ nalD_{T11N}$ | 16 | 2 |
| $dacB_{\Delta 13}\ nalD_{T158P}$ | 16 | 1 |
| $mpl_{insG}\ ampC_{P180L}$ | 1 | 1 |
| $mpl_{insG}\ ampC_{G242R}$ | 2 | 2 |
| $mpl_{insG}\ nalD_{T11N}$ | 2 | 2 |
| $mpl_{insG}\ nalD_{T158P}$ | 2 | 2 |
| $ampC_{P180L}\ nalD_{T11N}$ | 2 | 2 |
| $ampC_{P180L}\ nalD_{T158P}$ | 1 | 1 |
| $ampC_{G242R}\ nalD_{T11N}$ | 4 | 1 |
| $ampC_{G242R}\ nalD_{T158P}$ | 2 | 1 |
| $dacB_{\Delta 13}\ mpl_{insG}\ ampC_{G242R}$ | 64 | 0.5 |
| $dacB_{\Delta 13}\ mpl_{insG}\ nalD_{T11N}$ | 16 | 2 |
| $mpl_{insG}\ ampC_{G242R}\ nalD_{T11N}$ | 4 | 4 |

†MIC values are the median of at least three biological replicates, with up to three independently generated mutants. Values of individual experiments are in S4 and S5 Tables in S1 File. Clinical breakpoints for resistance are: ceftazidime, $\geq$ 32 mg/L; meropenem, $\geq$ 8 mg/L.

mutants *dacB ampC* and *dacB mpl* had higher ceftazidime MICs than those of bacteria with only a *dacB* mutation (Table 3). All other pairwise combinations of mutations did not increase the MIC above that of one of the single mutations. Furthermore, the MICs of bacteria engineered to contain combinations of three mutations were no higher than those of bacteria

**Table 4. Minimum inhibitory concentration for *P. aeruginosa* PAO1 transposon mutants.**

| Mutated gene | Ceftazidime MIC (mg/L)* | Meropenem MIC (mg/L) |
|---|---|---|
| Wild-type | 1 | 1 |
| *dacB* | 16 | 1 |
| *mpl* | 1 | 1 |
| *nalD* | 2 | 1.5 |
| *ampC* | 1.5 | 0.75 |
| *mexR* | 4 | 2 |

*MIC is the median of 4 biological replicates of strain PAO1 with transposon insertions in the genes shown [45]. Values of individual experiments are in S6 Table in S1 File.

**Table 5. Frequencies of predicted change-of-function variants in ceftazidime-sensitive and -resistant clinical isolates\*.**

| Protein | Sensitive isolates (n = 166) | Resistant isolates (n = 173) |
|---|---|---|
| AmpC | 54 (32.5%) | 67 (38.7%) |
| AmpR | 10 (6.0%) | 29 (16.8%) |
| DacB | 42 (25.3%) | 56 (32.4%) |
| MexB | 15 (9.0%) | 13 (7.5%) |
| MexR | 19 (11.4%) | 10 (5.8%) |
| Mpl | 10 (6.0%) | 39 (22.5%) |
| NalC | 4 (2.4%) | 6 (3.5%) |
| NalD | 18 (10.8%) | 27 (15.6%) |
| PhoQ | 16 (9.6%) | 17 (9.8%) |
| SltB1 | 4 (2.4%) | 4 (2.3%) |

\*Likely change-of-function sequence variants in the proteins shown were identified using Provean. The analysed genomes are listed in S7 Table in S1 File.

containing two mutations (Table 3). In comparison with mutants containing single mutations, there was little change to the MIC of meropenem when mutations were present in combination (Table 3).

## Prevalence of mutations in ceftazidime-resistant and -sensitive clinical isolates

The prevalence of mutations in resistance-associated genes that are likely to affect protein function in clinical isolates of *P. aeruginosa* is unknown. We therefore analysed the genome sequences of 173 ceftazidime-resistant and 166 ceftazidime-sensitive clinical isolates for sequence variants likely to the affect function of proteins identified in the experimental evolution study (Table 5). AmpC had the highest frequency of variants amongst both resistant (38.7%) and sensitive (32.5%) of isolates, likely reflecting its contribution to resistance to a range of ß-lactams in addition to ceftazidime, although the effects of sequence variants will be affected by levels of gene expression. PBP4 (*dacB*-encoded) function-altering sequence variants were present in almost a third of resistant isolates, but also about a quarter of sensitive ones. Variants in *mpl* were over-represented in resistant isolates (22.5%) compared to sensitive ones (6.0%). Variants in the other genes analysed were present in lower frequencies, in both resistant and sensitive isolates. Acquired ß-lactamases associated with ceftazidime resistance were not identified in any of the genomes.

## Discussion

Ceftazidime is a key antibiotic in the treatment of *P. aeruginosa* infections but a significant proportion of isolates are ceftazidime-resistant. Our results emphasise that ceftazidime resistance is multifactorial, involving multiple cellular processes. Mutations affecting different pathways in some but not all cases had additive effects on the MIC, with *dacB* mutations being primary contributors to reduced susceptibility and mutations in *ampC* and efflux regulators also playing a role.

Sequence variants likely to affect the function of PBP4, encoded by *dacB*, occurred at the highest frequency in experimentally evolved mutants. They also had the largest impact on MIC when engineered into *P. aeruginosa* and were present in ceftazidime-resistant isolates (Tables 1, 3 and 4). Collectively these findings emphasise that *dacB* mutations are a major

driver of ceftazidime resistance in *P. aeruginosa*. Inactivation of PBP4 results in accumulation of peptidoglycan-derived muropeptides that activate AmpR, the regulator of *ampC* expression, resulting in increased synthesis of AmpC ß-lactamase and a higher MIC for ceftazidime [23–26, 46]. However, the finding that PAO1 mutants containing only *dacB* mutations did not reach the threshold for resistance and the presence of predicted function-altering PBP4 sequence variants in ceftazidime-sensitive clinical isolates show that *dacB* mutations do not by themselves render *P. aeruginosa* ceftazidime-resistant. Mutations in *mpl* also arose frequently in the experimentally-evolved mutants as well as in previous studies [21, 22]. Although a mutation in *mpl* increased production of ß-lactamase [23], *mpl* mutations did not increase the MIC of strain PAO1 and *mpl* mutations were no more frequent in resistant than sensitive clinical isolates (Tables 3–5). The basis for the high frequency of *mpl* mutants in experimentally-evolved mutants, and why increased amount of ß-lactamase do not increase MIC, is therefore unclear. Mutations in *ampD* have also been associated with increased expression of *ampC* and increased MIC for ceftazidime in experimentally evolved *P. aeruginosa* [25, 42] but arose in only one mutant in this study, perhaps due to differences in the selection protocols used. Mutations in the *ampC* gene itself caused small increases in MIC, suggesting that they increase the activity of AmpC against ceftazidime and consistent with previous findings that *ampC* mutations can increase hydrolysis of ceftazidime and raise the MIC [20, 40].

ß-lactam effectiveness is also influenced by the rate of uptake of antibiotic into the periplasm and the rate of excretion by efflux pumps. Many of the experimentally evolved mutants had mutations that affect LPS synthesis suggesting that changes to LPS reduce entry of ceftazidime into the periplasm. The primary efflux pump for ceftazidime excretion is MexAB-OprM, expression of which is regulated by the MexR and NalCD proteins. Mutations in *nalC*, *nalD* and *mexR* all caused increases in MIC for ceftazidime, consistent with increased expression of *mexABoprM* in these mutants.

Although *dacB* mutations had the biggest impact on ceftazidime MIC, engineering a *dacB* mutant to contain additional mutations in *ampC* showed that coupling of alterations in AmpC with increased production of the enzyme had additive effects in increasing the MIC. However, although mutations in efflux pump regulators and in *dacB*/*ampC* reduce the intracellular concentrations of ß-lactams through different mechanisms, combining *nalD* mutations with *dacB* or *ampC* mutations did not further increase the MIC. These findings suggest that the MIC is not determined by a simple additive effect of mutations in different pathways.

Many of the experimentally-evolved mutants had higher MICs than those of the most ceftazidime-tolerant engineered mutants, and 11 of the mutants had higher MICs than the parental strains despite having no mutations in any of *dacB*, *mpl*, *ampC*, *nalC*, *nalD* or *mexR*. The majority of the ceftazidime-resistant clinical isolates did not have mutations likely to affect the function of *dacB*-encoded PBP4 (Table 5). These findings emphasise the contributions of mutations in other genes to the resistance phenotype as well as the complex and multifactorial nature of resistance.

Many of the experimentally-evolved mutants had mutations in genes encoding enzymes required for metabolic processes indicating that metabolic changes reduce the effectiveness of ceftazidime against *P. aeruginosa*. The wide range of affected pathways suggests that there was selection for mutations affecting metabolism *per se*, rather than individual pathways. It has recently become clear that changes to metabolic function can increase the ability of bacteria to tolerate antibiotics, especially when antibiotic exposure is interspersed with incubation in the absence of antibiotics as done here [47, 48]. It is notable that *P. aeruginosa* undergoes metabolic changes during chronic infection (reviewed in [49]) although the extent to which these changes influence antibiotic susceptibility has not generally been considered.

Thirteen of the 35 experimentally-evolved mutants also contained large (194–915 kb) deletions. Deletions have been reported previously in mutants evolved to be resistant to ceftazidime [21, 22] or meropenem [28, 42] although exactly how they contribute to increases in MIC is not known. All but two of the deletions include the *galU* gene that encodes an enzyme required for LPS core synthesis and has previously been associated with increased MIC for ceftazidime [23, 50]. Loss of a large number of genes is also likely to impact on cellular metabolism and it may be that the combination of metabolic changes and loss of *galU* reduces ceftazidime susceptibility to a greater extent than either one alone. Deletions also arise in isolates of *P. aeruginosa* from chronically-infected patients [28, 51–54] but whether they affect antibiotic resistance in these isolates has not been investigated. As well as increased MICs for ceftazidime, the experimentally evolved mutants had increased MICs of meropenem of between 2- and 32-fold relative to the parental strain. The experimentally-evolved mutants did not have mutations in genes commonly associated with meropenem resistance, notably *oprD* and *ftsI* [19], implying that cross-resistance to meropenem involves other mechanisms. Mutations in *mexR*, *nalC* and *nalD* caused a 2- to 4-fold increase in MIC (Tables 3 and 4), indicating that increased expression of the MexAB-OprM efflux pump played a partial role in increasing the MIC. Further increases in the MIC may be due to metabolic changes or changes to LPS.

Use of two different reference strains PAO1 and PA14 allowed us to compare the effects of strain-strain differences on evolution of ceftazidime resistance. As might be expected, many of the same cellular processes were affected in both strains during development of resistance. However, there were also some intriguing inter-strain differences. A higher proportion of PA14 mutants had mutations in efflux pump-related genes than the PAO1 mutants perhaps reflecting inter-strain differences in the contributions of efflux pumps to ceftazidime tolerance. Furthermore, a much higher proportion of PA14 mutants had mutations in *phoQ* and *clpA* although how mutations in these genes relate to ceftazidime susceptibility is not known. These inter-strain differences emphasise that ceftazidime resistance can involve strain-specific genome variation superimposed on species-wide resistance mechanisms. This is further emphasized by the occurrence of clinical isolates that are resistant to ceftazidime despite having no significant sequence variants likely to alter the function of PBP4. Characterisation of ceftazidime-resistant isolates of *P. aeruginosa* will shed further light on the extent to which the genetic basis of resistance varies between isolates. A more complete understanding of the genetic basis of ceftazidime resistance will also facilitate ongoing efforts to predict resistance phenotypes from the genome sequences of *P. aeruginosa* isolates (eg. [7, 8]).

## Conclusions

Mutations in *dacB* have the greatest effects on increasing tolerance of *P. aeruginosa* to ceftazidime but a *dacB* mutation does not by itself confer resistance in strain PAO1 or in clinical isolates of *P. aeruginosa*. Instead, our findings demonstrate that ceftazidime resistance is multifactorial, quantifying the additive effects of mutations in different genes, and can involve characterised antibiotic resistance mechanisms but also changes to LPS synthesis and metabolic functions. The additive effects of some but not all mutations and the occurrence of ceftazidime-resistant clinical isolates that apparently have functional PBP4, of ceftazidime-sensitive isolates in which PBP4 has likely function-altering variants, and of differences in the mutation profiles of experimentally evolved PAO1 and PA14 all emphasise the complex and multi-faceted nature of ceftazidime resistance. Nonetheless, quantifying the effects of different mutations on ceftazidime tolerance will facilitate ongoing efforts to predict antibiotic susceptibility from whole genome sequencing of infecting bacteria, an approach with potential to optimize treatment of infected patients.

## Supporting information

**S1 File.**
(XLSX)

## Acknowledgments

We gratefully acknowledge Dr Daniel Pletzer for providing the PAO1 transposon mutants.

## Author Contributions

**Conceptualization:** Scott C. Bell, Wayne M. Patrick, Craig Winstanley, Iain L. Lamont.

**Data curation:** Kay A. Ramsay, Samuel T. Wardell.

**Formal analysis:** Kay A. Ramsay, Samuel T. Wardell.

**Funding acquisition:** Scott C. Bell, Wayne M. Patrick, Craig Winstanley, Iain L. Lamont.

**Investigation:** Kay A. Ramsay, Attika Rehman, Samuel T. Wardell, Lois W. Martin.

**Project administration:** Iain L. Lamont.

**Supervision:** Iain L. Lamont.

**Validation:** Kay A. Ramsay, Attika Rehman, Samuel T. Wardell, Lois W. Martin.

**Visualization:** Kay A. Ramsay.

**Writing – original draft:** Kay A. Ramsay, Iain L. Lamont.

**Writing – review & editing:** Kay A. Ramsay, Attika Rehman, Samuel T. Wardell, Scott C. Bell, Wayne M. Patrick, Craig Winstanley, Iain L. Lamont.

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
