## [Decision Letter · Decision Letter 0]

3 Apr 2023

PONE-D-23-06436Ceftazidime resistance in Pseudomonas aeruginosa is multigenic and complexPLOS ONE

Dear Dr. Ramsay,

Thank you for submitting your manuscript to PLOS ONE. After careful consideration, we feel that it has merit but does not fully meet PLOS ONE’s publication criteria as it currently stands. Therefore, we invite you to submit a revised version of the manuscript that addresses the points raised during the review process.

We look forward to receiving your revised manuscript.

Kind regards,

Abdelwahab Omri, Pharm B, Ph.D, Laurentian University

Academic Editor

PLOS ONE

Journal Requirements:

Reviewers' comments:

Reviewer's Responses to Questions

**Comments to the Author**

1. Is the manuscript technically sound, and do the data support the conclusions?

Reviewer #1: Yes

Reviewer #2: Yes

2. Has the statistical analysis been performed appropriately and rigorously? 

Reviewer #1: N/A

Reviewer #2: Yes

3. Have the authors made all data underlying the findings in their manuscript fully available?

Reviewer #1: Yes

Reviewer #2: Yes

4. Is the manuscript presented in an intelligible fashion and written in standard English?

Reviewer #1: Yes

Reviewer #2: Yes

5. Review Comments to the Author

Reviewer #1: This manuscript by Ramsay and colleagues investigates how genetic mutations interact to confer resistance to ceftazidime. It is clearly written and logically presented; the experiments are technically sound. A few comments for the authors' consideration:

1. Is there enough information in the data presented in Table S2 and 3 to understand something about the order in which mutations might occur? Are there mutations that only occur conditioned on the acquisition of another mutation? For instance, it appears that in PA14 clpA mutations only occur if there is also a phoQ mutation, but not the other way around. This kind of analysis, if it is possible, would add to the robustness of the discussion.

2. Can the authors please discuss why there is almost no (sometimes a negative) change in meropenem resistance in the engineered PAO1 variants?

3. It's not at all clear what the transposon mutant table (Table 4) adds to the story. The authors have already gone through the trouble of making a variety of deletions and point mutants!

Reviewer #2: Useful study analyzing the effect of individual and combined mechanisms leading to ceftazidme resistance in P. aeruginosa obtained after in vitro evolution experiments using PAO1 and PA14 strains. There are however some points for the authors to consider:

1. Lines 48-49. There are other well stablished mechanisms leading to AmpC overexpression such as inactivation of AmpD or specif AmpR mutations.

2. Related to this, it is somewhat surprising that none of the evolved mutants showed AmpD mutations. It should be commented in the discussion.

3. Table 5. Difficult to interpret without expression data for ampC and mexB. A limitation statement should be added.

4. Lines 287-289. PAO1 dacB mutant is the only one showing clinical resistance levels

5. Lines 293-294 increase in ampC expression produced by mpl inactivation much lower than that produced by dacB or ampD inactivation.

6. Line 295-298. Usually AmpC mutations require AmpC overexpression to have a significant effect and therefore expected to add synergistic with dacB mutations. Frequently these AmpC mutations decrease MIC for carbapenems.

7. Lines 336-345. Usually these deletions show other typical characteristic such as brown pigment (hmgA deletion) and colistin (galU deletion) and aminoglycoside (meXY deletion) hypersusceptibility.

8. Some other expected mechanisms such as PBP3 mutations were not detected and might be discussed.

9. For discussion. A recent work determined a genomic score for predicting ceftazidime (and other antibiotics) resistance (Cortés-Lara et al CMI 2021).

6. PLOS authors have the option to publish the peer review history of their article (what does this mean?). If published, this will include your full peer review and any attached files.

Reviewer #1: **Yes: **Ajai Dandekar

Reviewer #2: No

---

## [Author Response · Author response to Decision Letter 0]

1 May 2023

Response to reviewers’ comments on manuscript PONE-D-23-06436

Thank you for providing the reviewers’ comments on our submitted manuscript. We are grateful to the reviewers for the time spent in considering our manuscript, and for their constructive suggestions on how to improve it. We have now adjusted the manuscript in light of the reviewers’ comments and to clarify issues that they have raised.

A point-by-point response to all of the reviewers’ comments on the earlier version of this manuscript is given below. Reviewers’ comments are shown in normal font, with our responses in italics. All references to line numbers in our responses refer to the numbering in the marked up revised manuscript. Please refer to the document PONE-D-23-06436_response to reviewers for detailed comments. 

We hope that our manuscript will now be acceptable for publication in PLoS One. 

We will look forward to hearing from you.

Kay A Ramsay, Corresponding author on behalf of all the authors.

---

## [Editor Report · Decision Letter 1]

3 May 2023

Ceftazidime resistance in *Pseudomonas aeruginosa* is multigenic and complex

PONE-D-23-06436R1

Dear Dr. Kay Ramsay,

We’re pleased to inform you that your manuscript has been judged scientifically suitable for publication and will be formally accepted for publication once it meets all outstanding technical requirements.

Kind regards,

Abdelwahab Omri, Pharm B, Ph.D, Laurentian University, Canada

Academic Editor

PLOS ONE

---

## [Editor Report · Acceptance letter]

8 May 2023

PONE-D-23-06436R1 

Ceftazidime resistance in *Pseudomonas aeruginosa* is multigenic and complex 

Dear Dr. Ramsay:

I'm pleased to inform you that your manuscript has been deemed suitable for publication in PLOS ONE. Congratulations! Your manuscript is now with our production department. 

Kind regards, 

on behalf of

Dr. Abdelwahab Omri 

Academic Editor

PLOS ONE